# An RNA Virome Analysis of the Pink-Winged Grasshopper *Atractomorpha sinensis*

**DOI:** 10.3390/insects14010009

**Published:** 2022-12-22

**Authors:** Yu-Juan He, Zhuang-Xin Ye, Chuan-Xi Zhang, Jun-Min Li, Jian-Ping Chen, Gang Lu

**Affiliations:** State Key Laboratory for Managing Biotic and Chemical Threats to the Quality and Safety of Agro-Products, Key Laboratory of Biotechnology in Plant Protection of MARA and Zhejiang Province, Institute of Plant Virology, Ningbo University, Ningbo 315211, China

**Keywords:** *Atractomorpha sinensis*, grasshopper, RNA virome, high-throughput sequencing, RNA interference pathway

## Abstract

**Simple Summary:**

The present work focused on the RNA virome analysis of an agricultural pest. In this study, the four novel RNA viruses in *A. sinensis* were reported. According to the phylogenetic analysis, they were classified as nege-like virus, iflavirus, chu-like virus, and ollusvirus. The complete genome sequence of four RNA viruses were obtained with RACE. Moreover, the analysis of virus-derived small interfering RNAs showed that most of the RNA viruses were targeted by the host antiviral RNA interference pathway.

**Abstract:**

A large number of RNA viruses have been discovered in most insect orders using high-throughput sequencing (HTS) and advanced bioinformatics methods. In this study, an RNA virome of the grasshopper was systematically identified in *Atractomorpha sinensis* (Orthoptera: Pyrgomorphidae), an important agricultural pest known as the pink-winged grasshopper. These insect viruses were classified as the nege-like virus, iflavirus, ollusvirus, and chu-like virus using HTS and phylogenetic analyses. Meanwhile, the full sequences of four novel RNA viruses were obtained with RACE and named Atractomorpha sinensis nege-like virus 1 (ASNV1), Atractomorpha sinensis iflavirus 1 (ASIV1), Atractomorpha sinensis ollusvirus 1 (ASOV1), and Atractomorpha sinensis chu-like virus 1 (ASCV1), respectively. Moreover, the analysis of virus-derived small interfering RNAs showed that most of the RNA viruses were targeted by the host antiviral RNA interference pathway. Moreover, our results provide a comprehensive analysis on the RNA virome of *A. sinensis*.

## 1. Introduction

With the aid of large-scale metatranscriptomic sequencing, an increasing number of novel RNA viruses have been discovered in insects, which can broaden our horizons of RNA viruses’ diversity [1,2]. These insect RNA viruses (IRVs) cannot infect vertebrates or plants and are present in many virus taxa, including Flaviviridae, Togaviridae, Reoviridae, Rhabdoviridae, Dicistroviridae, Iflaviridae, Aliusviridae, Chuviridae, and a proposed taxon “Negevirus” [3,4,5].

Hematophagous and sap-sucking insects play an important role in the transmission of arthropod-borne viruses (arboviruses). Previous studies have shown that several IRVs isolated from hematophagous insects interact with arboviruses and affect vector competence for arboviruses. For example, IRVs in the Flaviviridae modulate arboviruses’ replication and transmission in mosquitoes and sandflies [6,7,8]. Negeviruses inhibit the replication of several Alphaviruses during coinfection of mosquito cells [9]. For sap-sucking insects, including planthoppers, whiteflies, aphids, and leafhoppers, IRVs have been extensively investigated, and some IRVs appeared to be host-specific, indicating a long-term co-evolution between IRVs and host insects [10,11,12,13]. IRVs are thought to mediate sex ratio and increase pupal duration and fecundity in the host insect [14,15]. An analysis of viral genomic diversity and phylogeny has suggested that IRVs might be the ancestors of arboviruses and plant viruses [2,16].

*Atractomorpha sinensis*, commonly known as the pink-winged grasshopper (Orthoptera: Pyrgomorphidae) [17,18], is an important agricultural pest, directly feeding on leaves, buds, and tender stems of numerous cultivated plants in the families Leguminosae, Solanaceae, and Cruciferae. This species has caused severe damage to crop production in many countries including the Asia–Pacific region and North America [17]. In terms of virus transmission, grasshoppers have been considered as a vector to transmit vesicular stomatitis virus to hoofed animals (e.g., cattle and horse) [19,20]. Nevertheless, as a member of the grasshopper group, *A. sinensis* has not attracted wide attention to its outbreak and potential damage over the past several decades.

In this study, four novel RNA viruses were identified in *A. sinensis*. As revealed through analysis on sRNA data, RNAi-mediated antiviral immunity was active during the infection of these RNA viruses.

## 2. Materials and Methods

### 2.1. Sample Collection and Species Identification

A single adult pink-winged grasshopper was collected in the field of Huzhou, Zhejiang, China, in September 2019. This sample was delivered to our laboratory and stored at −80 °C until further analysis. Then, insect morphology was identified under a dissecting microscope and verified with Sanger sequencing using a pair of universal primers to amplify mitochondrial cytochrome c oxidase subunit 1 (COI) genes.

### 2.2. RNA Extraction and Sequencing

The single sample was homogenized in liquid nitrogen, and total RNA with rRNA was extracted using TRIzol reagent (Invitrogen, Carlsbad, CA, USA) following the manufacturer’s instructions. High throughput sequencing (HTS) and small RNA sequencing were conducted after evaluating the quality of total RNA with rRNA through Nanodrop (Thermo Scientific, MA, USA). The Illumina TruSeq Total RNA with rRNA Sample Preparation Kit was adopted to construct the cDNA library. A small RNA library was established using the Truseq Small RNA Library Preparation Kit (Illumina, San Diego, CA, USA) and sequenced based on the Illumina HiSeq 2500 platform, which resulted in 19 Mbp of raw data. Furthermore, raw RNA reads were subject to quality control, adapter trimming, and contig assembly using Trinity program (Version 2.8.5) [21] with default parameter settings.

### 2.3. Virus Genome Identification and Small RNA Analysis

To identify viral reads, the assembled RNA-seq contigs were queried against the non-redundant nucleotide (nt) and non-redundant protein (nr) sequence databases in NCBI using Blastn and Blastx algorithm, respectively. Furthermore, the candidate viral reads were compared against the Viral RefSeq database downloaded from GenBank in order to avoid false positive reads. E-value cut-off was set at 2 × 10^−20^ for each comparison. Based on blast hit results, unassembled overlaps and same viral reads were merged by using SeqMan program of DNAstar software package to form partial/complete virus genomes. All candidate viral reads were verified through RT-PCR following Sanger sequencing. The first strand of complementary DNA (cDNA) from 1000 ng of total RNA was synthesized with HiScript ^®^II Q RT SuperMix for PCR (+gDNA wiper) (Vazyme, Nanjing, China) in line with the manufacturer’s protocol. RT-PCR was performed in 10 µL-reaction agents containing 0.5 µL template cDNA, 5 µL of 2 × Phanta Max Buffer, 0.2 µL dNTP mix, 0.2 µL of Phanta Max Super-Fidelity DNA Polymerase (Vazyme, Nanjing, China), 0.2 µL of 1 µM forward and reverse primers, and 3.7 µL of ddH_2_O. The thermal cycling conditions were 95 °C for 5 min, 40 cycles of 95 °C for 30 s, 60 °C for 30 s, and 70 °C for 30 s. Abundance of each virus genome was estimated by mapping quality-checked raw reads to virus genomes with Bowtie 2 and Samtools programs [22,23]. Then, the coverage of the aligned reads was visualized using the Integrative Genomics Viewer (version 2.13.2) [24].

To obtain the full-length genome of the candidate virus, the rapid amplification of cDNA ends (RACE) assay was performed as described [25]. Briefly, 5′- and 3′-RACE cDNAs were synthesized using random primers or oligo (dT) primers present in SMARTer^®^ RACE 5′/3′ Kit (Takara, Kyoto, Japan). Both 5′ and 3′ PCR products were amplified with sequence-specific primers with nest-PCR and cloned into a pMD19-T clone vector individually (Takara, Kyoto, Japan) after Sanger sequencing, to obtain the complete virus genome sequence. The primers used for identification of virus-like sequences are listed in Appendix A.

To explore the small interfering RNA (siRNA) triggered by viruses in *A. sinensis*, the clean small RNA (sRNA) reads (18- to 30-nt) were mapped back to full-length virus genome sequences using Bowtie 2 with zero mismatches. In addition to that, virus-derived siRNAs were further vegetated using the Linux bash scripts and the custom perl scripts.

### 2.4. Virus Genome Annotation

The potential open reading frames (ORFs) in the virus genomes were predicted using the ORF Finder program in NCBI. The function annotation of each ORF was conducted through the NCBI Conserved Domain Database (https://www.ncbi.nlm.nih.gov/cdd/, accessed on 5 March 2022) and InterProScan Search (http://www.ebi.ac.uk/interpro/, accessed on 5 March 2022).

### 2.5. Phylogenetic Analyses

Phylogenetic analyses were carried out as described previously with modifications [25]. The amino acid sequence of the viral RNA-dependent RNA polymerase (RdRP) domain was aligned using MAFFT (v7.487) [26] with default parameters, while ambiguous portions of the alignment were trimmed by Gblock (0.91b) [27]. Phylogenies were improved after removing divergent and ambiguously aligned blocks from protein sequence alignments. Thereafter, the best-fit model of amino acid substitution was evaluated with ModelTest-NG [28]. Then, maximum-likelihood (ML) trees were obtained using RAxML-NG (v. 0.9.0) [29] with 1000 bootstrap replications and refined with iTOL [30]. The reference sequences of viral RdRP used for phylogenetic analyses are listed in Appendix A.

## 3. Results

### 3.1. Discovery of RNA Virus-Related Sequences in A. sinensis

To identify the RNA virus-related sequences in *A. sinensis*, the total RNA sample was subjected to RNA sequencing analysis using HTS. The results showed that 12.9 Gbp raw data (SRR21712770) were generated from the obtained cDNA library, which had deep sequencing on the Illumina HiSeq 4000 platform (150-bp paired-end reads) of Novogene. In total, 86,784,590 reads were generated and assembled de novo into 464,844 reads in the library. Through a Blastn search against the nucleotide database in NCBI, it was found that the insect COI sequence showed a high homology (99% identities) to that of *A. sinensis* (accession number: MW085548.1). In total, 17,356 virus-like reads were found in the library. After a BLASTn search against the NCBI nucleotide (nt) and the entire viral reference database, four novel RNA viruses were identified in the assembled contigs of A. sinensis, which were classified as nege-like virus, iflavirus, ollusvirus, and chu-like virus, respectively. The sequences of four viruses were uploaded to NCBI, and the accession numbers include OL672484, OL672485, OL672486, and OL672487. It was shown that three reads were almost identical to those of the Aphid lethal paralysis virus (ALPV), suggesting that these reads were derived from a strain of ALPV (Table 1 and Appendix A). All of the virus-like sequences were determined using RT-PCR with the specific primers and verified using Sanger sequencing (Appendix A).

### 3.2. Characterization of Atractomorpha sinensis Nege-Like Virus 1

The full-length sequence of Atractomorpha sinensis nege-like virus 1 (ASNV1) was 12,278 nt (excluding polyA tail), which contained a 69 nt 5′ untranslated region (UTR), a 457 nt 3′ UTR, and four predicted ORFs that further included ORF1 (70–8520 nt, encoding 2817 aa), ORF2 (8570–10,630 nt, encoding 687 aa), ORF3 (10,654–11,097 nt, encoding 148 aa), and ORF4 (11,137–11,820 nt, encoding 228 aa). ORF1 was featured by four conserved domains, namely, Vmethyltransf super family (vMet) domain (253–1089 nt), FtsJ-like methyltransferase (FtsJ) domain (2206–2775 nt), Viral_helicase1 (Hel) domain (5689–6483 nt), and RdRP_2 super family (RdRP) domain (6994–8448 nt) (Figure 1A), whereas ORF4 contained SP24 super family (SP24) domain (11,278–11,718 nt). In addition, ASNV1 replicated efficiently in *A. sinensis*, as evidenced by the highly abundant distribution of read coverage in the whole genome and the small RNAs with both positive and negative sense (Figure 1C).

The analysis of the phylogenetic relationship showed that ASNV1 was the cluster with Ganwon-do negev-like virus 1, together with Ingleside virus, Nephila clavipes virus 4, and Hubei virga-like virus 13, forming unclassified group1, showing a close relationship with the Kitaviridae (Figure 1B). Analyses of nucleotide and amino acid identity based on the RdRP sequences manifested that ASNV1 is related to other four viruses in Group 1, sharing 29–73.5% nt and 44.1–77% aa sequence identity (Appendix A).

The sRNA library was constructed to analyze virus-derived small RNA (vsiRNA) of ASNV1; as a result, ASNV1 triggered vsiRNA with peaks of 22 nt related to RNAi-mediated antiviral immunity (Figure 1C). The vsiRNA was generated from the whole genome of ASNV1, including untranslated regions and the asymmetric hotspots on both strands, implying that the host immune system might target these regions preferably (Figure 1E). Otherwise, the viral siRNAs of ASNV1 showed a strong A/U preference in their 5′-terminal nucleotides (Figure 1D), which is typical of vsiRNAs from various organisms, including insects.

### 3.3. Characterization of Atractomorpha sinensis Iflavirus 1

The full-length sequence of Atractomorpha sinensis iflavirus 1 (ASIV1) contains 9, 412 nt with the 791 nt 5′ UTR, the 118 nt 3′UTR, and a predicted ORF (792–9293 nt), where the ORF contained two rhv_like (rhv) domains (1827–2060 nt and 2631-2954 nt), the CRPV_capsid super family (CRPV) domain (3648–4304 nt), the Viral_helicase1 (Hel) domain (5208–5531 nt), and the RdRP domain (7761–9254 nt) (Figure 2A). Meanwhile, according to the relatively abundant distribution of the read’s coverage in the whole genome of iflavirus 1 and the small RNAs with both positive and negative sense, iflavirus 1 in *A. sinensis* replicated in an efficient way (Figure 2A).

Moreover, phylogenetic analysis results demonstrated that ASIV1 displayed a close relationship with another virus in Iflaviridae, especially Hubei coleoptera virus 1 and Tribolium castaneum iflavirus (Figure 2B). Furthermore, according to analyses on the conserved amino acid and the nucleotide sequence of RdRP domain, ASIV1 clustered closely with Tribolium castaneum iflavirus, with 34.4% aa and 41.3% nt sequence identification (Appendix A).

For vsiRNA, ASIV1 triggered vsiRNA with peaks of 22 nt (Figure 2C), which generated the untranslated regions and asymmetric hotspots on both strands from the whole genome of ASIV1 (Figure 2E). The viral siRNAs of ASNV1 had a strong A/U preference in their 5′-terminal nucleotides (Figure 2D).

### 3.4. Characterization of Atractomorpha sinensis Ollusvirus 1 and Atractomorpha sinensis Chu-Like Virus 1

The full-length sequences of Atractomorpha sinensis ollusvirus 1 (ASOV1) and Atractomorpha sinensis chu-like virus 1 (ASCV1) possess 14,715 nt and 7673 nt, accordingly. The sequence structures including an 85 nt 5′UTR, a 252 nt 3′UTR and three predicted ORFs were included in ASOV1, where the three predicted ORFs were ORF1 (86-3406 nt) encoding 1107 amino acids, ORF2 (4440-6805 nt) encoding 789 aa, and ORF3 (7039-14,463 nt) encoding 2475 aa. Meanwhile, the InterProScan was employed to predict the conserved domain of ASOV1. The ORF1 and ORF2 had no conserved domain, while ORF3 possessed three conserved domains, namely, RdRP domain (7069-10,206 nt), mRNA_cap domain (10,369-10,980 nt), and FtsJ-like methyltransferase (FtsJ) domain (13,153-13,416 nt) (Figure 3A). For ASCV1, the sequence consists of a 1799 nt 5′ UTR, a 1, 497 nt 3′ UTR, and three predicted ORFs that are ORF1 (1800-3518 nt), ORF2 (3491-4999 nt), and ORF3 (4888-6177 nt). Through the domain prediction, ORF1 has one conserved domain called RdRP (1806-3413 nt). ORF2 also possesses one conserved domain called mRNA_cap (3533-4279 nt), and one domain of ORF3 is called VP39 (5533-6102 nt) (Figure 3B). According to the relatively abundant distribution of the read’s coverage in the whole genome and small RNAs with both positive and negative sense, ASOV1 and ASCV1 replicated effectively in *A. sinensis* (Figure 3A,B).

Furthermore, the evolutionary relationship of ASOV1 and ASCV1 was explored with other viruses with phylogenetic analysis. ASOV1 had a close relationship with Culverton virus and Scaldis River bee virus, both of which were classified into the ollusvirus in the Aliusviridae with the aa sequence identity of 28.5–52.2%. ASCV1 clustered with Blattodean chu-related virus, which belonged to an unclassified group (Group 2) of Chuviridae, with 61.3% aa and 64.2% nt sequence identities (Figure 3C and Appendix A).

ASOV1 triggered the peaks of 22 nt (Figure 3D), and untranslated regions and asymmetric hotspots on both strands of the whole genome were observed in vsiRNA reads (Figure 3F), while viral siRNAs of ASOV1 exhibited a strong U preference in the 5′-terminal nucleotides (Figure 3E). However, ASCV1 triggered no peaks of 22 nt (Appendix A), and neither the untranslated regions or asymmetric hotspots on both strands of the whole genome were observed in vsiRNA reads (Appendix A).

## 4. Discussion

A large number of RNA viruses were discovered using HTS and advanced bioinformatics methods, thus, gaining a better understanding of insect viromes and viral evolution. Recently, numerous RNA viruses have been identified in multiple grasshopper species [34]. In this study, the four novel RNA viruses, namely, Atractomorpha sinensis nege-like virus 1 (ASNL1), Atractomorpha sinensis iflavirus 1 (ASIV1), Atractomorpha sinensis ollusvirus 1 (ASOV1), and Atractomorpha sinensis chu-like virus 1 (ASCV1) were found in *A. sinensis*. Negeviruses have a broad host range as a group of insect-specific viruses, including the orders of Coleoptera, Diptera, Hemiptera, Hymenoptera, Lepidoptera, Odonata, Orthoptera, and Thysanoptera [25]. In the meantime, negeviruses are characterized with a ssRNA (+) genome with poly(A) tails that are around 7000 nt to 10,000 nt in length and contain at least three ORFs [3,35]. Iflaviruses are the non-enveloped insect viruses that contain the ssRNA (+) genome with poly(A) tails at 3′UTR and encode one large polyprotein. For the polyprotein, the viral coat protein is located in an N-terminal domain and the non-structural proteins which are involved in replication and polyprotein processing in the C-terminal region [36]. In addition, the negative-sense RNA viruses, Chuviruses and Ollusviruses, belong to the Jingchuvirales order [37,38]. The genome of Chuviruses varies with unsegmented, bi-segmented, and circular forms, and most chuvirus genomes contain glycoprotein (G), nucleoprotein (N), and the large polymerase (L) genes. However, some genes are deleted during the evolution of Chuviruses, such as glycoprotein genes [39]. In our study, the genomic feature of ASNV1 was consistent with that of most negeviruses possessing four ORFs (Figure 1). Similar to other iflaviruses, ASIV1 contains one polyprotein, with the CRPV and Hel domain located at the N- and C- terminus, respectively (Figure 2). Moreover, the genome of ASCV1 encoded large polymerase (RdRP) and capsid proteins (VP39), but not G protein, which might be deleted during the virus–host co-evolution process (Figure 3).

As potential biocontrol agents, ISVs have attracted more attention due to their effects on vectoring ability. In addition, they impede some arboviruses’ infections in vivo and in vitro in mosquitos, such as Nhumirim virus and Zika virus, participate in reducing West Nile virus transmission when co-infecting the mosquito vector [4], and are suppressed by Menghaic rhabdovirus and Shinobi tetravirus in the *A. albopictus* C6/36 cell line [40]. Otherwise, it is known that ISVs regulate the innate immune pathway in some insects, such as mosquitoes, to interfere with the replication by decreasing vector competence [4,40,41]. In our study, the antiviral RNA interference pathway of *A. sinensis* responded strongly during the infection of three RNA viruses.

In conclusion, ASNV1, ASIV1, ASOV1, and ASCV1 were identified in *A. sinensis* and classified as nege-like virus, iflavirus, ollusvirus, and chu-like virus according to a phylogenetic analysis. Except for ASCV1, other RNA viruses were targeted by the antiviral RNA interference pathway in *A. sinensis*.

## Figures and Tables

**Figure 1 insects-14-00009-f001:**
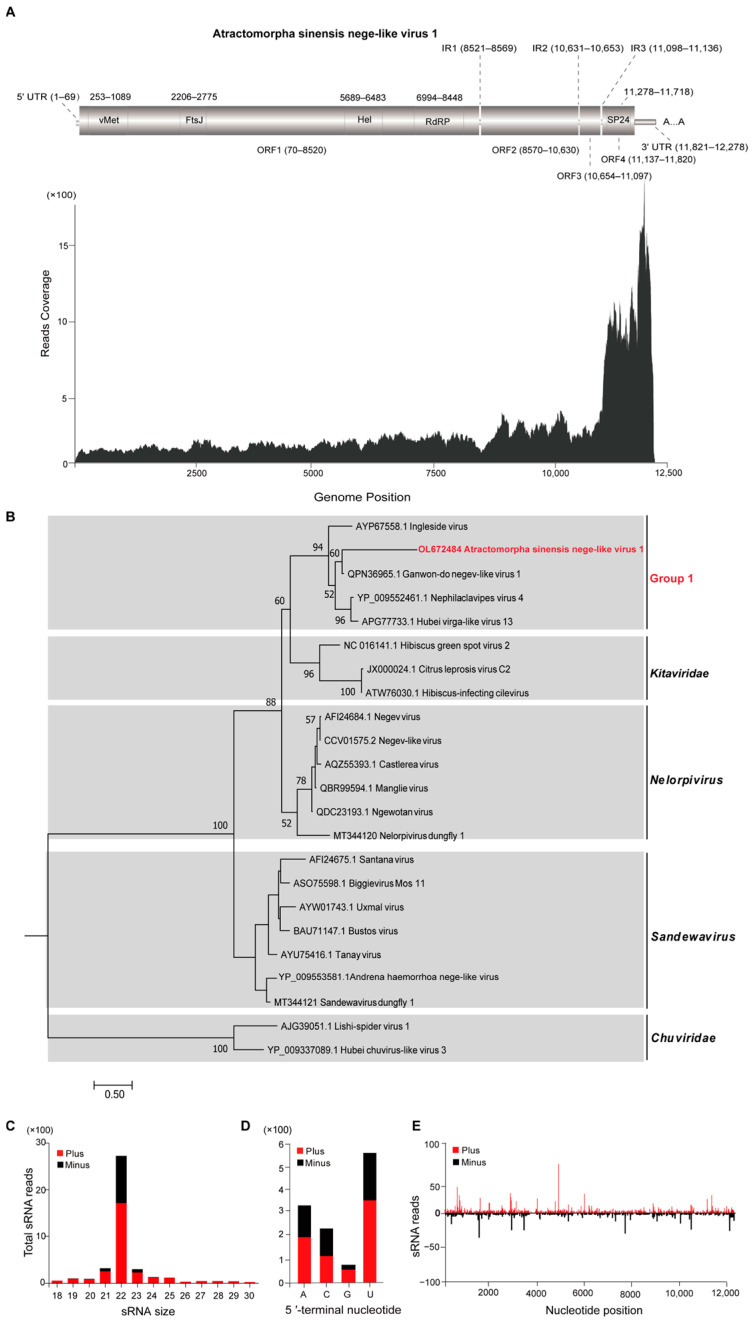
Atractomorpha sinensis nege-like virus. (**A**) Genetic structure and transcriptome raw read coverage; vMet, viral methyltransferase domain; FtsJ, Ribosomal RNA methyltransferase; Hel, RNA virus helicase core domain; RdRP, RNA-dependent RNA polymerase domain; SP24, a putative virion membrane protein. (**B**) Phylogenetic analysis of ASNV1 and other negeviruses based on the conserved RdRP amino acid sequences. Bootstrap values are displayed over each node of the tree (when > 50), and scale bars represent percentage divergence, while the ASNV1 identified in this study is indicated with red fonts. (**C**) Size distribution of ASNV1 sRNAs. (**D**) Preference of 5′-terminal nucleotide. (**E**) Distribution of ASNLV1-derived sRNA along the corresponding viral genome.

**Figure 2 insects-14-00009-f002:**
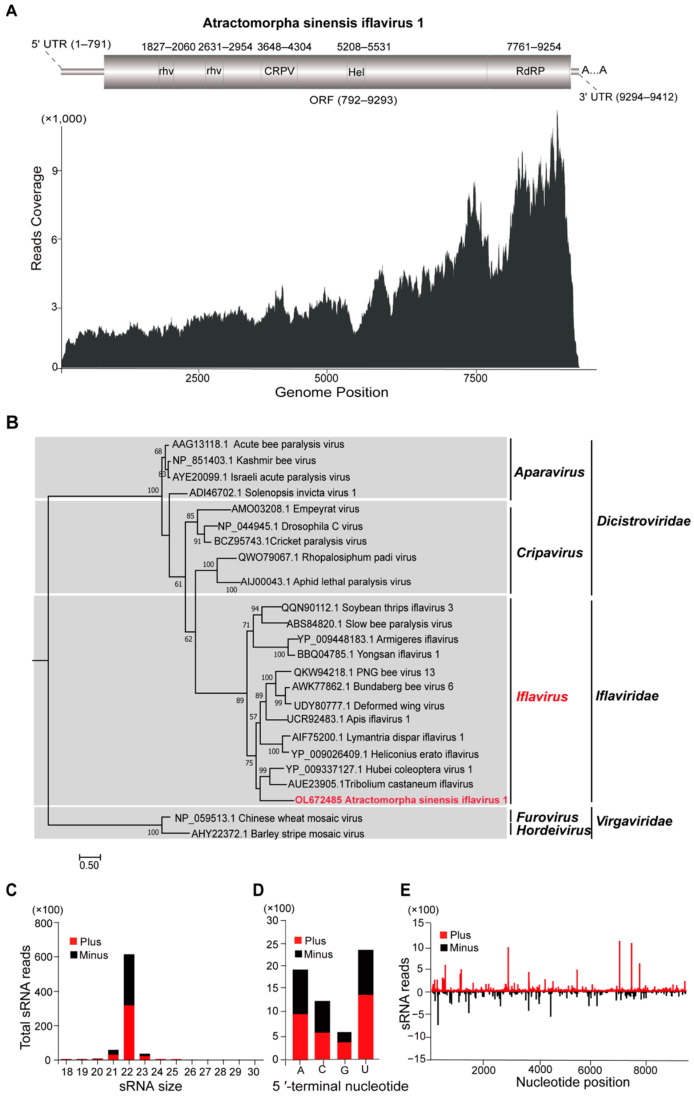
Atractomorpha sinensis iflavirus 1. (**A**) Genetic structure and transcriptome raw read coverage; rhv, picornavirus-like capsid protein; CRPV, CRPV_capsid super family, cricket paralysis virus; Hel, RNA virus helicase core domain; RdRP, RNA-dependent RNA polymerase domain. (**B**) Phylogenetic analysis of ASIV1 and previously reported viruses in the iflavirus based on the conserved RdRP amino acid sequences. Bootstrap values are placed over each node of the tree (when > 50). Scale bars represent percentage divergence, while the ASIV1 identified in this study is indicated with red fonts. (**C**) Size distribution of ASIV1 sRNAs and 5′-terminal nucleotides. (**D**) Preference of 5′-terminal nucleotide. (**E**) Distribution of ASIV1 derived sRNA along the corresponding viral genome.

**Figure 3 insects-14-00009-f003:**
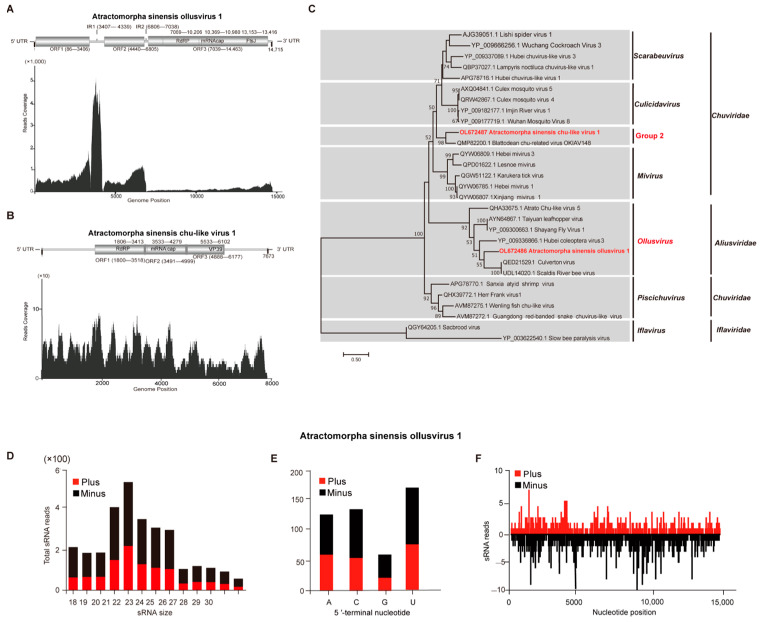
Atractomorpha sinensis ollusvirus 1 and Atractomorpha sinensis chu-like virus 1. (**A**) Genetic structure and transcriptome raw read coverage of Atractomorpha sinensis ollusvirus 1 (ASOV1); FtsJ, Ribosomal RNA methyltransferase; RdRP, RNA-dependent RNA polymerase domain. (**B**) Genetic structure and transcriptome raw read coverage of Atractomorpha sinensis chu like virus 1 (ASCV1); VP39, Capsid Proteins, RNA-dependent RNA polymerase domain. (**C**) Phylogenetic analysis of ASOV1 and ASCV1 and previously reported viruses in the Chuviridae, Aliusviridae, and Iflaviridae based on the conserved amino acid sequences. Bootstrap values are placed over each node of the tree (when > 50). Scale bars represent percentage divergence, and ASOV1 and ASCV1 identified in this study are indicated with red fonts. (**D**) Size distribution of ASOV1 sRNAs. (**E**) Preference of 5′-terminal nucleotide. (**F**) Size distribution of ASOV1-derived sRNA along the corresponding viral genome.

**Table 1 insects-14-00009-t001:** The detailed messages about four novel viruses identified in *Atractomorpha sinensis*.

Tentative Virus Names	NCBI Accession	Length (nt)	Classification	Abundance	E-Value	RdRP Protein Identities	Top BlastP Hit Virus	Virus Reference
Atractomorpha sinensis nege-like virus 1 (ASNLV1)	OL672484	12,278	Nege-like virus	112.2	6 × 10^−110^	42.13%	Ganwon-do negev-like virus 1	[31]
Atractomorpha sinensis iflavirus 1 (ASIV1)	OL672485	9412	Iflavirus	1689.9	2 × 10^−129^	45.63%	Hubei coleoptera virus 1	[22]
Atractomorpha sinensis ollusvirus 1 (ASOV1)	OL672486	14,715	Ollusvirus	251.5	0.0	50.84%	Culverton virus	[32]
Atractomorpha sinensis chu-like virus 1 (ASCLV1)	OL672487	7673	Unclassified chuviridae	4.8	3 × 10^−165^	63.96%	Blattodean chu-related virus OKIAV148	[33]

## Data Availability

The data presented in this study are available within the article and Appendix A.

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
