# Peer review of "An RNA Virome Analysis of the Pink-Winged Grasshopper Atractomorpha sinensis"

_insects, 2022, doi:10.3390/insects14010009_

Round 1
Reviewer 1 Report
Overall, I found this manuscript to be well-written and containing information useful to both the entomological and virological communities. I endorse this manuscript for publication with minor revisions as indicated in the reviewed version of the manuscript that I've attached. I've included 82 edits/comments that should be addressed before final acceptance. The majority are simply grammatical issues with some occasional comments, all of which, in my opinion, will make this work even stronger.

Author Response
Overall, I found this manuscript to be well-written and containing information useful to both the entomological and virological communities. I endorse this manuscript for publication with minor revisions as indicated in the reviewed version of the manuscript that I've attached. I've included 82 edits/comments that should be addressed before final acceptance. The majority are simply grammatical issues with some occasional comments, all of which, in my opinion, will make this work even stronger.
Response: Thank you for your great comments and suggestions. And some revisions have been made in revised manuscript and highlighted.
- “Atractomorpha sinensis nege-like virus 1 (ASNV1), Atractomorpha sinensis iflavirus 1 (ASIV1), Atractomorpha sinensis ollusvirus 1 (ASOV1) and Atractomorpha sinensis chu-like virus 1 (ASCV1)”. Italicize, *repeat for others - I understand it's part of an acronym, but they're still species names, so I would argue they still need to stand out as such.
Response: Thank you for this suggestion. However, it doesn't need to be italicized when a species name is attached to a virus in standard virus classification.
- “A single adult pink winged grasshopper was collected in the field of Huzhou”. why only one? Is that standard? Or is there a reason only one was used?
Response: Thank you for this suggestion. Because only one grasshopper was collected in field and we wanted to figure out how many types of viruses exist in this grasshopper.
- “Then, insect morphology was identified under a dissecting microscope and verified by Sanger sequencing using a pair of universal primers to amplify mitochondrial cytochrome c oxidase subunit 1 (COI) genes”. what is meant here? Do you mean that morphology was used to verify the identify of the species and then verified again using Sanger sequencing? If so, then this needs to be clarified.
Response: Thanks for pointing this out. A dissecting microscope was used to make a preliminary determination of which insect, and Sanger sequencing was performed to clear the precise species name.
- “In addition to that, virus-derived siRNAs were further unvegetated using the Linux bash scripts and the custom perl scripts”. This doesn't seem correct in this context since this means "to be free of vegetation"
Response: Thanks for pointing this mistake. We have revised the description in the paper.

Reviewer 2 Report
The manuscript is a first step towards characterizing the virome of A. sinensis and with that contributes to our understanding of viral diversity and evolution. The description of data is structured in a way that each result part and corresponding figure has the same lay-out, which makes it easy to follow. Overall, the manuscript has merit in that it describes new viral sequences from an insect that has not been analyzed before. However, only a single grasshopper analyzed, sampling size is limited and no information is available about the prevalence of these viruses.
Q1: There is a lack of references or accession date (for online programs) for some software used such as Trinity program.
Q2: Line 111 “Systematic biology, 2007, 56(4): 111 564-577.” should be deleted.
Q3: In the result part, viruses labeled in the red font in the phylogenetic tree should be indicated.
Q4: The four viruses are found in just one grasshopper. Was only one grasshopper analyzed? If material for other grasshoppers is available, it would be nice to test the presence of these viruses to get some indication of the prevalence of the virus in the population.
Q5: The author should do some functional research for these viruses.
Author Response
The manuscript is a first step towards characterizing the virome of A. sinensis and with that contributes to our understanding of viral diversity and evolution. The description of data is structured in a way that each result part and corresponding figure has the same lay-out, which makes it easy to follow. Overall, the manuscript has merit in that it describes new viral sequences from an insect that has not been analyzed before. However, only a single grasshopper analyzed, sampling size is limited and no information is available about the prevalence of these viruses.
Response: Thank you for your great comments and suggestions.
Q1: There is a lack of references or accession date (for online programs) for some software used such as Trinity program.
Response: Thanks for pointing this out. We have added it.
Q2: Line 111 “Systematic biology, 2007, 56(4): 111 564-577.” should be deleted.
Response: Thank you for this suggestion. We have deleted it.
Q3: In the result part, viruses labeled in the red font in the phylogenetic tree should be indicated.
Response: Thank you for this advice. We have revised it.
Q4: The four viruses are found in just one grasshopper. Was only one grasshopper analyzed? If material for other grasshoppers is available, it would be nice to test the presence of these viruses to get some indication of the prevalence of the virus in the population.
Response: Thank you for this suggestion. Because only one grasshopper was collected in field, we are unable to test the rate of viral infection.
Q5: The author should do some functional research for these viruses.
Response: Thank you for great advice. Because the grasshopper populations amounts was too little to be not suitable for function study.
